# Pitch Improvement in Attentional Blink: A Study across Audiovisual Asymmetries

**DOI:** 10.3390/bs14020145

**Published:** 2024-02-19

**Authors:** Haoping Yang, Biye Cai, Wenjie Tan, Li Luo, Zonghao Zhang

**Affiliations:** 1School of Physical Education and Sports Science, Soochow University, Suzhou 215021, China; lingyangking@126.com (H.Y.); 20234006005@stu.suda.edu.cn (B.C.); 2Suzhou Cognitive Psychology Co-Operative Society, Soochow University, Suzhou 215021, China; sptanwenjie@mail.scut.edu.cn; 3Department of Physical Education, South China University of Technology, Guangzhou 518100, China

**Keywords:** attentional blink, audiovisual integration, cross modal, pitch perception

## Abstract

Attentional blink (AB) is a phenomenon in which the perception of a second target is impaired when it appears within 200–500 ms after the first target. Sound affects an AB and is accompanied by the appearance of an asymmetry during audiovisual integration, but it is not known whether this is related to the tonal representation of sound. The aim of the present study was to investigate the effect of audiovisual asymmetry on attentional blink and whether the presentation of pitch improves the ability to detect a target during an AB that is accompanied by audiovisual asymmetry. The results showed that as the lag increased, the subject’s target recognition improved and the pitch produced further improvements. These improvements exhibited a significant asymmetry across the audiovisual channel. Our findings could contribute to better utilizations of audiovisual integration resources to improve attentional transients and auditory recognition decline, which could be useful in areas such as driving and education.

## 1. Introduction

Attentional blink is a phenomenon of attention blindness in the temporal dimension that occurs due to human limitations in attention allocation [1]. Specifically, in the rapid serial visual presentation (RSVP) stream, when subjects need to attend to two targets (target 1, T1; target 2, T2), if the time interval between the two targets is 200–500 ms and there are multiple consecutive and non-repeating interfering targets in between, subjects’ recognition of T2 will be significantly affected by the temporal factor. ABs reveal the temporal limitation of attentional resources and contribute to the understanding of the dynamic allocation mechanisms of attention. The study of ABs provides insights into the understanding of attention, working memory, and perceptual recognition processes. The AB effect may explain some information processing errors, such as lapses in driving and monitoring tasks. The results of AB studies can help to improve the design of human–computer interactions and to reduce the number of errors caused by information overload.

There are many hypotheses to explain this phenomenon, with the attentional gate model suggesting that it is caused by the failure of T2 to open the door when T1 passes through the attentional gate [1]. The two-stage model suggests that it is the limitation of attentional resources at a given time that leads to an AB [2]. This coincides with the temporary loss of control hypothesis where human attention is out of control in this time frame [3]. On the other hand, the central interference model points to the limitation of cognitive resources which interferes with memory processes during the extraction process to cause the AB [4,5]. This is further supported by the delayed engagement of attention hypothesis, which suggests that this is a dynamic process and that the occurrence of an AB is related to working memory [6]. These ideas were integrated into the bottleneck theory of attention, which attributes the creation of an AB to a bottleneck, which also provides a theoretical reference for improving ABs [7,8]. The elimination of ABs is currently an important topic in the research on attention. A growing body of research has found that an AB is improved when auditory stimulation accompanies the experiment [9,10,11,12,13,14]. However, in focusing on the auditory improvement of ABs, these studies failed to further explore the effects of auditory stimulus material representations on ABs. 

Many studies have confirmed that sound can be used to improve attention. Early studies have shown that sound improves ABs independent of the frequency of the auditory occurrences [9,12]. The reason for bimodal pitches not affecting visual recognition may be related to the presence of a correlation between the processing mode and ceiling effects. Similarly, the semantic coherence of auditory stimuli failed to significantly affect ABs [9,11]. These studies are based on “bottom-up” processing. In other words, when subjects change their processing mode, the recognition of the visual target may be affected by auditory factors. Pitch is an important element in the composition of sound. People perceive different pitches differently. For example, in everyday life, people are more likely to perceive higher pitched sounds; higher frequency sounds are perceived as brighter and sharper, and are therefore more likely to be noticed. In the classroom, teachers may raise the pitch to emphasize key knowledge in the textbook when they are talking about key content [15]. These cases show that changes in pitch can help improve attention.

Processing modality is a major factor influencing the perception of cross-visual and audiovisual attention. Different processing styles influence the exploration of attentional mechanisms [6,16,17]. The limited capacity of the resources needed for a distributive task of visual attention can be improved with audiovisual stimuli [3]. The mechanism underlying this improvement is worth exploring. Some studies have suggested that ABs are improved by auditory stimuli due to an alertness effect, i.e., the presence of auditory sounds activates early attentional resources. However, the effect of vigilance was ruled out by a comparison of audiovisual interleaving experiments, in which the early versus lagged emergence of sounds failed to significantly affect T2 discrimination. In the exploration of the relationship between auditory and visual influences, there is research that suggests that this is due to the fact that T2 is reinforced due to the accompaniment of sound only at T2, but there is no difference in the recognition of T2 when compared to a sound condition where all visual stimuli are accompanied by a sound [11,12]. Another view is that the visual improvement comes from the integration of vision and hearing, which is based on the reinforcement of vision through auditory activation. Many views consider this to be an automated process. Because of this, few studies have further explored how and where these automated processes arise. Different cognitive resources and pathways exist for both the visual and auditory systems. In tasks perceived across audiovisual channels, when asked to report with an auditory stimulus (target 3, T3), the recognition of auditory information is influenced by the sound and by where the auditory stimulus appears. This suggests that auditory stimuli are also cognitively loaded and need to occupy certain cognitive resources, including attentional resources and working memory. This view is consistent with the delayed engagement of attention hypothesis.

Evaluations of audiovisual integration are usually based on conditions in which audiovisuals are presented simultaneously, as opposed to conditions in which only visuals or only audio stimuli are presented. When changes in visual recognition do not coincide with changes in auditory recognition, this produces audiovisual asymmetry. Several studies have shown that when visual and auditory stimuli are presented simultaneously and require a discrimination between visual and auditory information, visual tasks are prioritized for debriefing and visual information tends to dominate the integration outcome [18]. A higher degree of reporting occurs when visual and auditory information are matched [19]. Similarly different time-invariant conditions also affect visual and auditory discrimination [11]. These studies have explored audiovisual integration in terms of its perceptual dimensions. However, the process of attention is complex and is related to different stages of attention and the working memory in addition to perception. Studying asymmetries in audiovisual integration can help to reveal the adaptive priorities of different senses in evolution and cognition. Differences between the perceptual channels can be clarified and cognitive models can be refined. Further research could provide ideas for more natural multimodal interactions.

It is not clear whether different pitches have different effects on improving ABs, or whether the portion of visual attention that is reinforced during audiovisual integration comes from auditory sources. In the present study, we increased the information representation of auditory stimuli by using three pitched sounds. By changing the processing mode to increase the auditory load, it can be further clarified whether this audiovisual integration comes from the borrowing of auditory resources. The purpose of this study was to explore the influence of audiovisual integration on ABs and whether the presentation of pitch in the process of an audiovisual integration of an AB will raise the detection target and improve the AB. Therefore, the present study hypothesized that (1) pitch and lag affect T2 and T3 recognition, and that (2) AB is improved under audiovisual conditions, which is due to the sound being delivered. This research helps to further clarify how auditory changes can be utilized to improve ABs and increase attentional capacities, to clarify the possible triggering scenarios of audiovisual asymmetry phenomena, and to explore and understand the mechanisms of sensory interactions.

## 2. Method

### 2.1. Participants

This study was a 2 × 4 experimental design. We estimated the sample size using G*Power 3.1.9.2 [20]. With an estimated effect size of f = 0.25, alpha = 0.05, and power = 0.9, G*Power suggested a sample size of 30 participants. In the actual recruitment, all subjects were recruited through an online platform using posters in the surrounding community. 

A total of 35 college students (19 females, M = 20.61, SD = 1.644) were included in the experiment. All subjects were right-handed and had normal or corrected-to-normal vision and normal hearing. There was no history of any psychiatric illness or disease. Three of the subjects had their data excluded. Two of the subjects were unable to accurately discriminate sound stimuli and were excluded, i.e., they provided too few correct auditory target reports. The other subject was excluded for providing too few correct visual target reports and was a non-blinker in the visual target recognition [21]. 

This study was a sub-study of another NSF-funded study. All subjects signed an informed consent form before the experiment and were paid after the completion of the experiment. All experiments complied with the Declaration of Helsinki, and the experiments were approved by the Ethics Committee of Soochow University (ESCU-2023000209).

### 2.2. Apparatus, Stimuli, and Experimental Setup

The visual stimuli were presented on a 19-inch monitor (DELLE2316HF) with a resolution of 1280 × 1024 and a refresh rate of 60 Hz. The background color of the screen was grey during presentation (RGB: 196, 196, 196). Subjects were asked to hold their eyes level with the center of the screen at a distance of approximately 80 cm.

All experiments were programmed through MATLAB 2019a and Psychtoolbox 3.1.2 [22] and the data were collected using MATLAB 2019a. All visual stimuli in the experiments consisted of numbers and letters [23]. Among them, numbers between 0 and 9 constituted distractors and were presented in an alternating sequence of odd and even numbers. Odd and even numbers were alternated to avoid consecutive occurrences of the same number. Capital letters were used as the first target (T1) and the second target (T2). Confusing letters (e.g., Z and 2, O and 0, I and 1, etc.) were not included. The final letters E, G, J, K, V, R, and B were used for T1 and U, T, P, A, Z, N, and Y were used for T2. Prior to conducting the experiment, the subjects only knew that the target stimuli for T1 and T2 were letters, but not the specifics of T1 and T2 [11].

The auditory material consisted of pure pitches of B in different keys, with an octave difference between the bass B4 (494 Hz), alto B5 (988 Hz), and treble B6 (1976 Hz). The loudness was consistent (75 dB) and playback time was 75 ms. All subjects heard a continuous and randomized string of audio stimuli prior to the experiment, and each sound was played for 75 ms and then there was a 25 ms interval until the next sound was played. All subjects included in the experiment reported being able to perceive audible distinctions between different pitches.

### 2.3. Experimental Procedures and Design

This experiment used a 2 (T2 presentation: presentation of letters vs. presentation of blank) × 4 (sound: bass accompaniment for T2 vs. alto accompaniment for T2 vs. treble accompaniment for T2 vs. no sound) × 2 (lag: lag3 vs. lag8) within-subject design. In total, there were 14 conditions (we eliminated the presentation of the blank and no sound conditions from our behavioral experiments), 36 trials per condition, and 504 trials (see Table 1). 

VA denotes the condition in which the audiovisual stimuli appeared synchronously. Only_V denotes that only visual stimuli appeared at T2 (no auditory stimuli). Only_A denotes that only audio stimuli appeared at T2 (no auditory stimuli). At the beginning of the experiment, the subjects were required to gaze at a gaze point (a fixation cross) at the center of the screen. An RSVP stream was presented on the screen after 1000 milliseconds. During an RSVP stream, 23 visual stimuli were presented sequentially on the screen. Each RSVP stream consisted of 21 number interference stimuli and 2 letter targets (T1, T2). Each stimulus consisted of a 50 ms presentation of the visual object (number or letter) with a 50 ms presentation of a blank screen. The blank screen was presented to keep similarities between consecutively presented visual stimuli from influencing visual judgments. For example, 3 and 8 would produce similar percepts when presented consecutively [11]. T1 appeared randomly at positions 6 through 13 in the RSVP stream to prevent the subjects from producing rhythms. Lag denotes the lag position of T2 relative to T1, i.e., lag3 indicates that T2 appeared at the 3rd stimulus position after the presentation of T1, and lag8 indicates that T2 appeared at the 8th stimulus position after the presentation of T1. The presentation of T2 was accompanied by a sound stimulus (T3). When all the stimuli were presented in full, the subjects were required to report answers for three questions on the keyboard: what T1 and T2 were, and whether the sound of T3 was a treble, a midrange, or a bass. Figure 1 shows a sample RSVP experiment. All reports were recorded via keyboard keys, and subjects were asked to respond as accurately as possible during the task. The experiment only recorded the percentage of correct results reported for each target.

### 2.4. Data Analysis

Behavioral data were recorded using MATLAB where T2T1 denotes the number of times T2 was correctly identified in an RSVP stream only if both T2 and T1 were reported correctly. T3T2 denotes the number of times T3 was correctly identified in an RSVP stream only if both T3 and T2T1 were reported correctly. T3T1 denotes the number of times T3 and T1 were correctly identified when T2 did not present a letter (T3T1). All data were tested for normality and analyzed for the effect of the lag and sound factors on the T2T1, T3T2, and T3T1. Descriptive results are shown in parentheses as means and standard errors.

Further comparisons of the effects of audiovisual asymmetry were made via subtraction [24]. The audiovisual asymmetry is reflected by the change in the effect of audiovisual presentation on vision (AVPV) and the effect of audiovisual presentation on audition (AVPA). The AVPV is calculated by subtracting the T2T1 from the audiovisual condition (VA_T2T1) from the T2T1 from the visual-only condition (Only_V_T2T1) and the AVPA is calculated by subtracting the T3 from the audiovisual condition (T3T2) from the T3 from the visual only condition (T3T1). This is calculated as follows: AVPV = VA_T2T1 − Only_V_T2T1.
AVPA = T3T2 − T3T1.

All data were subjected to a repeated measures ANOVA after satisfying the tests for sample independence, normality, and a chi-square test. We conducted a two-way repeated measures analysis of variance with lag and pitch as independent variables and T2T1, T3T1, AVPV, and AVPA as dependent variables. In the presence of interaction effects, a post hoc analysis using the least significant difference (LSD) method was performed for multiple comparisons.

## 3. Results

### 3.1. T2T1 Correct Rate

The results indicated a significant effect of lag (*F* (1, 31) = 18.19, *p* < 0.001, *η_P_*^2^ = 0.394) while the sound effects were not significant (*p* > 0.05). The sound and lag interaction effect was significant (*F* (2, 62) = 3.718, *p* = 0.017, *η_P_*^2^ = 0.107). A further simple effect analysis showed that under the bass accompaniment for T2 conditions, the subjects in the lag3 condition gave significantly fewer correct answers than in the lag8 condition (*p* < 0.001, *Cohen’s d* = −0.254, 95% CI [−0.112, −0.04]). In the alto accompaniment for T2 conditions, the subjects in the lag3 condition gave significantly fewer correct responses than those in the lag8 condition (*p* = 0.039, *Cohen’s d* = −0.109, 95% CI [−0.064, −0.02]). There was no significant difference between the lag3 and lag8 conditions under the treble accompaniment for T2 conditions (*p* > 0.05). In the no sound with T2 conditions, the subjects in the lag3 condition gave significantly fewer correct answers than those in the lag8 condition (*p* < 0.001, *Cohen’s d* = −0.253, 95% CI [−0.115, −0.036]) (see Figure 2a). Post hoc comparisons showed that in the lag8 condition, the subjects’ recognition of T2 with the treble accompaniment was lower than with the bass accompaniment (*p* = 0.031, *Cohen’s d* = −0.078, 95% CI [−0.045, −0.002]) and no sound accompaniment (*p* = 0.017, *Cohen’s d* = −0.099, 95% CI [−0.057, −0.006]) (see Figure 2b).

Attention was influenced by lag and pitch factors, exhibiting an AB effect. The magnitude of the AB phenomenon diminished with an elevated pitch. The elevation of pitch in the lag8 condition led instead to a decrease in T2 recognition.

### 3.2. T3T2 Correct Rate

The results showed a significant main effect of lag (*F* (1, 31) = 12.01, *p* = 0.002, *η_P_*^2^ = 0.279), with the effect of the lag3 condition (0.524, 0.051) being less than that of the lag8 condition (0.553, 0.051). The sound effects were also significant (*F* (2, 62) = 3.431, *p* = 0.039, *η_P_*^2^ = 0.1). The pairwise comparisons reveal that in the alto condition (0.504, 0.053) results reported by subjects were worse than in the bass condition (0.563, 0.048) (*p* = 0.042, *Cohen’s d* = −0.203, 95% CI [−0.115, −0.002]). The sound and lag interaction was not significant (see Figure 3).

Auditory discrimination was affected by lag and pitch. Specifically, the larger the lag the better its auditory discrimination. For pitch, the alto condition showed a lower level of recognition, while the treble vs. bass condition showed a higher level of recognition.

### 3.3. Comparison of the Changes in AVPV and AVPA

The results showed that the main effect of lag with the AV_Effect was not significant. Lag interacted significantly with the AV_Effect (*F* (1, 32) = 7.89, *p* = 0.009, *η_P_*^2^ = 0.203) (see Figure 4a). In lag8 conditions, the AVPV (0.103, 0.047) was greater than the AVPA (−0.017, 0.1) (*p* = 0.011, *Cohen’s d* = 0.272, 95% CI [0.029, 0.21]) (see Figure 4b). 

These results suggest that there is an asymmetry in the integration process across audiovisual channels and that this manifestation of asymmetry is different under different lag conditions. Again, this suggests that lag correspondence plays a crucial role in multisensory processing.

## 4. Discussion

### 4.1. Pitch and Lag Affect T2 Recognition

Consistent with previous findings, an accompanying sound improved ABs [11,12,14,25]. However, this improvement did not eliminate the ABs. This may be due to the fact that processing changes are associated with an enhanced auditory load. This corroborates the relationship between ABs and working memory in the central interference model hypothesis [4,5], indicating that deepening ABs is a top-down processing approach [26]. 

There is a uniqueness in the sound, but after changing the processing, subjects will need to perceive both auditory and visual information with T2, and the process of storing them in working memory may increase the cognitive load, interfering with the retention of visual information. The auditory reinforcement of vision is an automated process [27] that occurs in the early stages of attention and perceptual processing. Pitch’s reinforcement of visual attention may be more intuitively represented by before-and-after comparisons in real life. For example, when someone is emphasizing something, they produce a change in pitch. Thus, a change in pitch cannot directly affect visual information, but it requires a baseline of sorts for comparison. The reason for the non-significant pitch under the lag3 conditions can be explained by the two-stage model [2]. During this crowded passage, subjects were only able to perceive visual and auditory stimuli, but were unable to process them deeply. The cognitive resources that may be required to co-occupy attention under lag3 conditions resulted in the pitch not acting as a salient enhancer.

A trend is presented in Figure 2a, where the magnitude of the AB phenomenon decreased with increasing pitch. This reflects a correlation between the effect of changes in pitch and visual recognition. Bass pitches are less easily perceived when perceiving sounds [28], leading to a poor improvement in visual attention using bass pitches. Figure 2b shows that the elevation in pitch in the lag8 condition led instead to a decrease in T2 recognition. This phenomenon highlights the possibility of resource sharing and switching between the perception of visual and auditory stimuli. In other words, the decrease in visual stimulus recognition under the lag8 condition with the treble accompaniment for T2 was due to the fact that the auditory stimulus was allocated more attentional resources during the non-bottleneck period. 

### 4.2. Pitch and Lag Affect T3 Recognition

The lag and sound effects were significant, which suggests that auditory information equally occupies attentional resources, which is consistent with the visual pathway. The perceptual processing of audiovisual information occurs in parallel, and a person can attend to both visual and auditory information at the same time [29,30,31]. Pitch perception may belong to the early part of attention and may not occupy working memory resources [32]. This could explain why the AB phenomenon reappeared after a shift in processing.

In terms of discrimination, subjects need deep attentional engagement and some memory resources to discriminate auditory stimuli. Interestingly, at the end of the study, the subjects were rarely able to report all the tones, and only some of the population with specialized musical training were able to report them accurately. This suggests the possibility that not everyone is able to identify specific pitches accurately and quickly. In other words, the process of subjects identifying pitches in an experiment is a process that requires more cognitive resources. This study of pitch recognition requires the auditory system to analyze pitch changes, which is more resource-intensive than sound detection alone [33]. Pitch recognition requires the pitch information to be maintained and compared in working memory, which requires additional memory resource support [34]. The audiovisual integration theory proposes that sensory processing requires the sharing of limited attentional resources. Increasing the difficulty of an auditory task increases its resource requirements [35]. The significance of the lag effect verified the possibility that the same AB was present for the auditory task discrimination in the cross-channel experiments [36]. In the actual study, there was a tendency for the sound to rise in all conditions, suggesting to some extent that T3 in the VA condition may also produce a transient deafferentation [37].

Bass sounds were weakly perceived by the subjects, but they were not difficult to discriminate, and instead, showed a higher degree of reporting under the lag8 conditions. In contrast, the subjects had more difficulties discriminating the midrange portion. This corroborates the point made above that more cognitive resources are needed to discriminate pitches [38,39]. The process of pitch discrimination occupies memory resources [39,40]. Pitch discrimination is a process that involves the brain’s processing and memory of sounds, and requires the brain to remember and compare previously heard sounds in order to recognize different pitches. This pattern of discrimination is influenced by pitch differences.

Correct T3 reports were associated with memory, which is consistent with the central interference model view. Working memory plays a key role in the short-term retention and integration of perceptual information. Our results further support the central interference model hypothesis [4,41].

The stimulus representations of sound did not produce an interaction with lag. This suggests that the effects of pitch and lag factors on attention transients are independent of each other, possibly because they have different processing pathways in the brain. Pitch typically involves the auditory cortex and related music processing areas, whereas the lag factor may be more associated with the time perception and cognitive control areas, such as the prefrontal cortex [42,43]. Thus, even when faced with a distracting situation, the brain may process the two factors independently without them appearing to influence each other. In conclusion, by comparing the T2T1 and T3T2 in different lag and pitch conditions, it can be further inferred that the effect of hearing on vision is unidirectional, but that vision and hearing are perceived in parallel. 

For auditory recognition, the stimulus representation of the sound did not interact with the lag, but the phenomenon present in visual recognition validates the prioritization of visual processing [18]. In RSVP tasks, visual stimuli may be more likely to capture and occupy attentional resources due to the task requirements and stimulus characteristics [19]. The absence of an interaction in auditory tasks may be due to the different types of tasks and ways of allocating attention. This difference may reflect the effect of different task types on perceptual channel interactions and that a relationship exists with the properties of the auditory stimulus. In visual recognition tasks, sound stimuli may cause a shift in attention or interference, which may affect the subject’s performance in the visual task. 

### 4.3. The Difference between the AVPV and AVPA Reflects the Asymmetry in Audiovisual Integration

In the position of T2, the simultaneous perception of vision and hearing in the T2 position is a dual task. It is not possible to conclude whether shared attention exists across audiovisual channels. This is because it is not part of the alerting and orienting part of attention [11,14], but it may be present in the higher processing stages of attention. In the results of the AVPV, it can be seen that the recognition of vision is instead improved in the dual task. This has been interpreted in some studies to be a result of audiovisual integration [12]. There is some debate about this point of inquiry, starting with the fact that some studies have argued that the process of audiovisual integration is automated [16]. However, the underlying premise of automation is that it does not require advanced processing [16]. Prior studies in experiments designed for stimulus conditions that are typically audio-visually synchronized do not need to report auditory, only visual information. Thus, perceiving auditory stimuli may exist only at the primary processing stage. Subsequent studies have found that the processing modality also affects audiovisual integration [11,16], but whether it enhances or inhibits it has not been standardized. Positive AVPV results have been reported to demonstrate that audiovisual integration is also present in “processing mode” or “dual-task” designs.

We found that this seemed to be related to the pitch level by controlling the pitch (Figure 2a). A comparison of the difference between lag3 and lag8 shows that the ABs seem to disappear as the pitch changes. However, in the actual statistics, there is a decreasing trend at lag8 and an increasing trend at lag3 as the pitch increases. This phenomenon we believe may be due to the fact that the total amount of cognitive resources is fixed [44]. If this view is valid, it can be inferred that there is a possibility for mutual transformation between audiovisual resources, and that this transformation process is most likely influenced by sound factors.

The visual improvement increased with increasing lag, but the auditory improvement decreased with increasing lag. Under lag3 conditions, both the visual and auditory information showed an enhancement effect, while the auditory information under lag 8 conditions showed a decrease. Under lag3 conditions, the VA condition showed a significant improvement in T2 reporting in the Only_V condition, and also produced a significant enhancement in the task of reporting auditory stimuli compared to the Only_A condition. This suggests that the bottleneck can be significantly expanded during the lag3 bottleneck phase under audiovisual synchronization, expanding the cognitive resources in this time frame. In other words, the integration across audiovisual channels activated additional cognitive resources during this time frame [44]. This additional activated cognitive resource may belong to a portion of the cognitive resources shared between the visual and auditory perception channels.

Similarly, this finding suggests that audiovisual integration can promote the brain’s efficiency and accuracy in processing information, especially when the time interval is short. This may be due to the fact that audiovisual integration can be complementary between visual and auditory channels [45]. Under lag8 conditions, the VA condition showed a significant improvement in T2 reporting over the Only_V condition, but produced a decrease in the reporting of auditory stimuli compared to the Only_A condition. The visual stimulus reporting under lag8 conditions demonstrated high-quality audiovisual integration. The accompaniment with auditory stimuli significantly improved visual recognition. However, in contrast, the recognition of auditory stimuli did not produce similar integration effects to what it did under lag3 conditions.

Auditory recognition under lag8 conditions decreased. An explanation, via the bottleneck theory, may be that more cognitive resources were readily available under lag8 conditions compared to the bottleneck period [7], and that these cognitive resources prioritized the processing of the visual information in the time-series task (the T2 recognition results were presented first relative to the T3 recognition results) [16]. This resulted in T2 producing a higher level of recognition. When T2 occupied more cognitive resources, there were fewer resources left for T3, thus producing fewer reports. This can be explained through the temporary loss of the control hypothesis which proposed that there may be a resource bias, and that attention is preferentially tilted towards T2 perception when perceiving T2 and T3 at the same time [3]. However, regarding the absence of this phenomenon under the lag3 conditions, it is more likely that it is difficult to tilt and shift cognitive resources at this point in time due to the fact that the resources in the bottleneck occupy the entire space of the bottleneck.

The integration between the audiovisual channels is not symmetrical. This asymmetry is reflected under the lag3 conditions, where the improvement in auditory recognition in the VA condition did not result in the same degree of improvement in visual recognition. This is also reflected under lag8 conditions, where significant improvements in visual recognition in the audiovisual condition were matched by significant decreases in auditory recognition. This asymmetry may be explained by the theory of biological information processing which posits that information from different perceptual channels is integrated and processed in the brain to produce an integrated perception of external stimuli. This integration may be modulated by the weights and effects of inputs from different perceptual channels, and this modulation may be asymmetric, resulting in asymmetric integration across audiovisual channels [46,47]. Alternatively, there are neuroscientific theories that suggest that there are specific networks of neurons in the brain that process information from different perceptual channels and that these neuronal networks may exhibit asymmetric activity and interactions during integration [48]. However, all these theories need further experiments to confirm them.

The present study also has limitations. Firstly, the sample size was small (although our sample size is consistent with the calculations of the sample estimation method we used), and thus, the generalizing of our findings should be done cautiously and allow for replication with a larger number of participants. Secondly, our cross-sectional design does not allow for inferring causality and requires changes in the experimental design to validate the results. Last but not least, our study focused only on behavioral results, but more electrophysiological, imaging, and other more in-depth methods should be adopted in subsequent studies.

## 5. Conclusions

This study explored the effects of pitch and lag factors on visual ABs and auditory recognition, and their derived asymmetry in audiovisual integration. Lag factors are the main factors in visual and auditory recognition. In visual recognition tasks, pitch factors can interact with lag factors to affect the perception of visual targets. In auditory recognition tasks, both lag and pitch can have a major effect on auditory recognition. In comparing the audiovisual integration task with the visual- and auditory-only tasks, inconsistent asymmetries were found for audiovisual integration at different temporal locations. However, both showed an enhanced recognition of visual factors. This finding could help people reduce ABs and improve their visual recognition, especially in real-life scenarios, such as driving, learning, and other domains.

## Figures and Tables

**Figure 1 behavsci-14-00145-f001:**
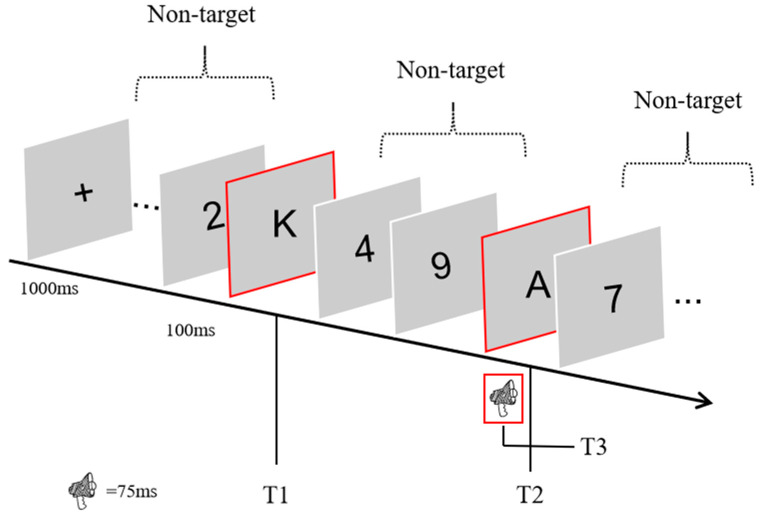
An RSVP experiment. After subjects had watched and listened to all of the audiovisual material, they were required to report T1 (‘K’), T2 (‘A’), and T3 (the pitch of the sound that appeared at the same time as T2) using the keyboard. T1, T2 & T3 are marked by red box.

**Figure 2 behavsci-14-00145-f002:**
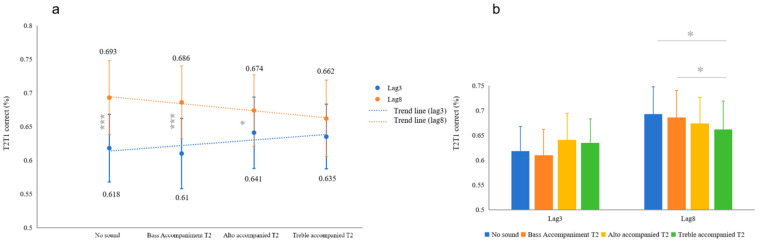
T2T1 results under different sound and lag conditions and trends. (**a**) The AB decreases as the pitch rises. (**b**) Comparison of differences between sound conditions. * *p* < 0.05; *** *p* < 0.001. Error bars represent the standard error.

**Figure 3 behavsci-14-00145-f003:**
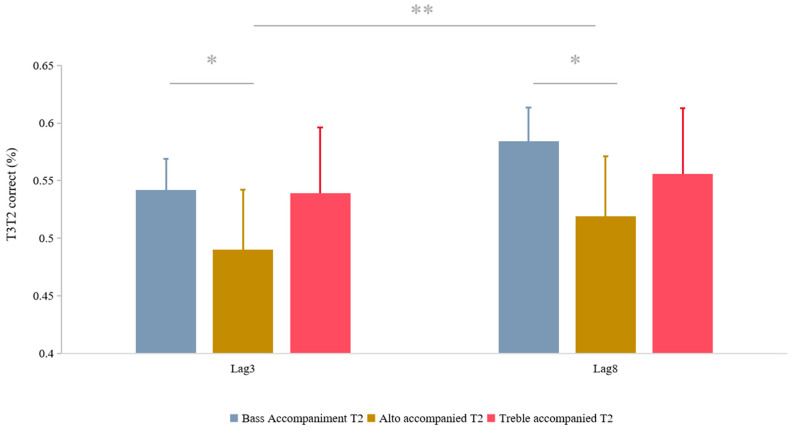
T3T2 results under different sound and lag conditions and trends. * *p* < 0.05; ** *p* < 0.01. Error bars represent the standard error.

**Figure 4 behavsci-14-00145-f004:**
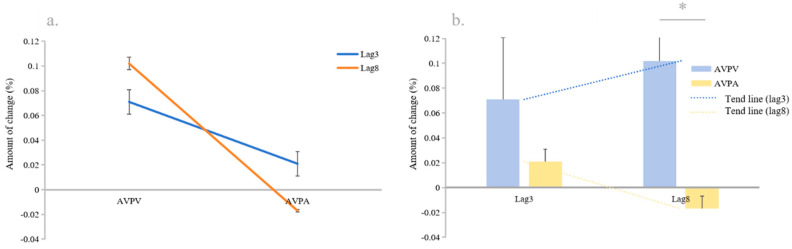
Comparison of the changes in T2 and T3. (**a**) Difference between AVPV and AVPA under different lag conditions. (**b**) Trend of AVPV vs. AVPA. * *p* < 0.05. Error bars represent the standard error.

**Table 1 behavsci-14-00145-t001:** Different experimental conditions of the RSVP experiment.

Visual and Audio Conditions	Factor (Level)	T1 Presentation and Identification	T2T1 Presentation and Identification	T3T2 Presentation and Identification
VA	Lag (2); sound (3)	Letter; report the corresponding letter on the keyboard	Letter; report the corresponding letter on the keyboard	Randomization of sounds; identification of high, medium, and low pitches
Only_V	Lag (2)	Letter; report the corresponding letter on the keyboard	Letter; report the corresponding letter on the keyboard	No sound; it is possible to press the space bar, but correctness is not recorded
Only_A	Lag (2); sound (3)	Letter; report the corresponding letter on the keyboard	Blank; it is possible to press the space bar, but correctness is not recorded	Randomization of sounds; identification of high, medium, and low pitches

## Data Availability

The data that support the findings of this study are available from the corresponding authors (Z.Z. or L.L.) upon reasonable request.

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
