# Peer review of "Pitch Improvement in Attentional Blink: A Study across Audiovisual Asymmetries"

_behavsci, 2024, doi:10.3390/bs14020145_

Round 1

Reviewer 1 Report

Comments and Suggestions for Authors

Dear Authors,

The present research article, entitled “Tone improvement in Attentional blink, a study across audiovisual asymmetries”, aims to investigate the effect of audiovisual asymmetry on attentional blinking and whether the presentation of tones improves the ability to detect a target during AB and is accompanied by an audiovisual asymmetry.

The main strength of this manuscript is that it seeks to clarify how auditory changes can be utilized to improve AB and increase attentional capacity.  In addition, to clarify the possible triggering scenarios of audiovisual asymmetry phenomena and to explore and understand the mechanisms of sensory interaction.

In general, I believe that the topic and approach of this article is timely and of interest to the readers of Behavioral Sciences.  However, I believe that some issues should be included to improve the quality of the manuscript.

Abstract:

Briefly describe the procedure carried out.  It is true that this part should be brief in the summary but I consider that “Thirty-five subjects were included in this study.  Differences in visual and auditory 16 recognition were compared by controlling for sound variables (three tones vs. no sound condition)”, is not enough to understand what the study consists of.

Introduction

I think it would be convenient to describe the objective at the end of the introduction.  The hypothesis is detailed but the objective only appears in the abstract.  It would be convenient to put it next to the hypothesis.

Methods

What were the inclusion criteria?  In order to understand the characteristics of the population used in the study, it would be very useful and advisable to detail the inclusion criteria.

Best regards.

Author Response

Reviewer 1

  1. Comment: Briefly describe the procedure carried out. It is true that this part should be brief in the summary but I consider that “Thirty-five subjects were included in this study. Differences in visual and auditory 16 recognition were compared by controlling for sound variables (three tones vs. no sound condition)”, is not enough to understand what the study consists of.

Response: We thank the reviewer for the careful consideration. We have revised it as follows: “Thirty-five subjects were included in this study. The differences in visual and auditory recognition in an audiovisual task versus a single-channel task were compared using lag versus pitch as the independent variable. (see pages 1, Abstract).

  1. Comment: I think it would be convenient to describe the objective at the end of the introduction. The hypothesis is detailed but the objective only appears in the abstract. It would be convenient to put it next to the hypothesis.

Response: Thank you for your insightful opinion. We have revised it as follows: “The purpose of this study is to explore the influence of audiovisual integration on AB and whether the presentation of pitch in the process of audiovisual integration of AB will raise the detection target and improve AB. (see pages 3, lines 115).

  1. Comment: What were the inclusion criteria?In order to understand the characteristics of the population used in the study, it would be very useful and advisable to detail the inclusion criteria.

Response: Thank you for your insightful opinion. We have detailed the inclusion criteria as follows: “A total of 35 college students (19 females, Mage = 20.61, SD = 1.644) were included in the experiment. All subjects were right-handed and had normal or corrected-to-normal vision and normal hearing. There was no history of any psychiatric illness or disease.” (see pages 3, lines 131).

Reviewer 2 Report

Comments and Suggestions for Authors

The authors investigate the interaction between visual and auditory signals on the phenomenon of attentional blinking (AB), specifically the extent to which the auditory signal improves the ability to detect a visual target during AB.

The findings of the paper seem very interesting and relevant, but it is nearly impossible for me to judge upon their validity, because the language of the manuscript is confusing and makes it difficult to understand what is going on. The methods and the results are not clearly described and the structure of the manuscript should be improved.

Here are some examples to illustrate my criticism:

  1. Abstract:

    1. "The results show that as lag increases the subject's target recognition improves and the voice produces further changes on top of it." What "voice"? The auditory signal? Or: "This improvement exhibited a significant asymmetry across the audiovisual channel when compared after subtraction from the baseline condition." The improvement exhibited an asymmetry? What is subtracted from what? What is compared to what? The sentence is unintelligible.

  2. Introduction:

    1. P2 L54ff: "However, in focusing on the auditory improvement of AB these studies failed to further explore the effects of auditory stimulus material representations on AB. However previous focuses on the sound improvement AB have overlooked the issue of sound characterisation." What is the contrast of the second "however"? Is this not repeating the same statement?

    2. In multiple places throughout the manuscript "tone" is used. What exactly is meant by "tone"? Is it not "pitch", which is also used at multiple places, but seemingly meaning something else than "tone"? For example, on P2 L58ff we read: "People perceive different tones differently. For example, in everyday life, people are more likely to be attracted to sharper sounds." What is a "sharp sound" in this context, and why are people "attracted" to it? This does not make much sense to me. And further we read: "In the classroom, the teacher may raise the pitch or make the voice lower when he/she is talking about the key points." A "lower" voice usually means a softer, quieter voice, but is this what the authors want to say here? They continue: "This shows that tone helps to improve visual perception." I do not see the connection here. Why does being "attracted" to a "sharper sound", or raising the pitch in the classroom support the conclusion that tone helps to "improve" visual perception? The authors go on: "Many studies have confirmed that looking at sound can improve vision." How could one possibly "look at sound"? And what type of "sound", is it music, or noise, or spoken words, and what is the connection to "tone", which is the actual topic of this paragraph?

    3. P2 L78ff: "Some studies have suggested that it is the uniqueness of the sound that improves AB, but when all stimuli are accompanied by sound AB does not differ from the accompanying sound at T2 only." AB is the phenomenon of attentional blink, which is in a totally different category than sound, so it is impossible that AB "differs" from the accompanying sound.

    4. P2 L91: "This leads to the phenomenon of asymmetry in audiovisual integration." What does "This" refer to?

    5. P2 L93ff: "In particular, the priority of visual information in audiovisual simultaneous judgement tasks [18]. The effect of visual quantitative information on audiovisual matching tasks [19]. Similarly affected by the time perception [11]." These are not full sentences and their meaning in the given context is unclear.

    6. P2 L98: "perceptual perception" is a funny tautology that seems out of place here.

    7. P3 L109f: "the improvement of sound for AB is due to the occupancy of auditory resources during recognition or recognition." It is certainly not the sound that is improved, but rather the attentional blink (AB) is improved, caused by the sound being delivered.

  3. Methods:

    1. Where do the authors get the estimated effect size of F=0.3?

    2. It is reported that "Two subjects failed to discriminate between auditory stimuli and one subject was excluded for low accuracy. In other words, non-blinkers were excluded" Was the auditory accuracy determined beforehand or after examining the results? What exactly was the criterion of performing too poorly? And I do not see why people that perform poorly on discriminating auditory stimuli are "non-blinkers." Intuitively, a "non-blinker" would be someone that shows no attentional blink, which, however, seems unrelated to not being able to distinguish between stimuli.

    3. P3 L138: "collat" should mean "collected".

    4. P3 L144: "was used for T1" should be "were used for T1".

    5. P3 L147ff: What does "si" mean? E.g. "494hz, si in C major", or "988hz, si in C ascending". And it is impossible that a single pitch can be related to an entire chord (C major, and what is "C ascending", do they mean "C augmented"?). A chord consists of at least three differently pitched notes. If there are only two notes, we have an interval, and a single note is just a single note.

    6. Experimental procedure: Why is there a presentation of "blank" in the visual stimuli, and a presentation of "no sound"?

    7. Table 1: What is "Trail"? What are "Corresponding letters of the report"? What is meant by the statement: "It is possible to press the SPACE BAR, but correctness is not recorded"?

    8. P4 L162ff: "[...] 23 visual stimuli were presented sequentially on the screen. Each stimulus was presented for 100 ms, with 50 ms of visual stimulus presentation followed by 50 ms of blank screen presentation." This does not make sense to me. There are 23 visual(!) stimuli that are followed by (another?) visual stimulus and a blank screen (which is, for that matter, also a visual stimulus)? In the next sentence, what can possibly be meant by "changes in visual stimuli were not obvious during visual perception"? And how can all these statements be reconciled with the statement that "When the presentation of the first stimulus ended, the next stimulus was presented immediately"? What happened to the blank screen in between? And how can this in turn be reconciled with "In this study the stimulus onset asynchrony (SOA) was only 100 ms"? Isn't it that the SOA is usually a variable time interval between the onset of subsequent stimuli? Here, it is fixed to 100 ms? And if so, how to understand the next sentence: "This is represented by the lag, which is the time interval of T2 relative to T1 when the SOA of T1 and T2 is 300 ms, or lag3"? And right after this sentence we learn that "Similarly, the other condition is lag8, which is the time interval of T2 relative to T1". But before it was lag3, not lag8? And after that we read the incomplete sentence: "21 number interference stimuli with 2 letter targets (T1, T2)." I am completely lost at this point.

    9. When the authors speak of "reports" do they possibly mean "responses"?

    10. How can it be that "subjects were required to respond accurately during the response" (P4 L175)?

    11. What is "T2 correctness with T1 correctness (T2T1), T3 correctness with T2T1 correctness (T3T2), ..."?

    12. P5 L189: "Effect of audiovisual presentation on vision (AVPV) = T2T1 in audiovisual condition (VA_T2T1) - T2T1 in visual-only condition (only_V_T2T1). " What do these equations and abbreviations represent? These are all incomplete sentences with no apparent meaning!

  4. Results:

    1. P5 L195f: "A two-way repeated measures ANOVA was conducted with T2T1 in the audiovisual condition as the dependent variable." This does not belong in the Results section but definitely in the Methiods section. And it has to be a bit more elaborate there. What are the two dependent variables (factors) of the two-way ANOVA? Have the requirements for an ANOVA been checked? Why is it a repeated-measures ANOVA and what is the temporal factor in this context?

    2. If and when there is a significant interaction effect between factors, the corresponding main effects must be taken with caution, and may be even meaningless. There is no discussion of this issue.

    3. "lag3 was significantly lower than lag8" Not the conditions themselves were lower but the subject's performance under these conditions.

    4. Why is there (apparently) no post-hoc comparison after the ANOVA of the T2T1 lag?

    5. Figure 2:

      1. What do the connection lines between the values represent? How come they are even curved in a rather weird way? They are indicated in the legend as "Tend lines". Do the authors mean "trend lines"? But even so, they are drawn between distinct categories, so there actually is no meaning to values in between these categories.

      2. What do the error bars represent?

      3. I assume the asterisks between two values with error bars on top of each other indicate a significant difference. Why is there no significant difference between the last two values, even though the errors bars do not overlap?

      4. Why did the authors choose such a rather unorthodox graphic representation and not a representation as in Figure 3? The data have a similar structure.

      5. P6 L220f: "Tonal stimuli, on the other hand, showed relatively centred sounds that was approximately difficult to discriminate." What does this sentence mean?

      6. Title of subsection 3.3: "Comparison of the amount of change in T2 and T3." What change is meant here?

      7. Also the description of the two-way repeated measures ANOVA belongs in the Methods section, with an appropriate and more elaborate explanation.

    6. General criticism:

      1. The fact that the auditory stimulus is presented at the same time as the second visual stimulus makes the discrimination tasks with respect to these two stimuli of different modality a dual-task scenario. Thus, there is a shared attention between the two discrimination tasks, which hinders a clear interpretation with respect to the actual aim of the study: to find out to what extent sound improves visual discrimination. In the experimental setting designed by the authors, the auditory stimulus draws attention away from the visual discrimination task, because the subject has to discriminate not only the visual stimuli but also the auditory stimuli. While this is interesting in itself, the discussion of this crucial point is very sparse and confusing. For example, on P8 L300, the authors write: "The stimulus representations of sound did not produce an interaction with lag. This suggests that their effects on attention are independent of each other." On the one hand, this is a very strong conclusion that is not in line with established views. Secondly, the authors found that "The sound and lag interaction effect was significant" (P5 L198f), which seems to directly contradict their statement that the stimulus representations of sound did not produce an interaction with the lag between the visual stimuli.

    In conclusion, the manuscript suffers from an unclear description of the methods and a confusing presentation and discussion of the results. Until these issues are adequately addressed, I cannot recommend publication of the paper.

Comments on the Quality of English Language

As stated in my main report, the language of the paper must be improved, as it hinders the understanding of the methods and the results. 

Author Response

Reviewer 2

Thank you for the valuable comments. We have revised the manuscript carefully as you suggested.

Abstract

  1. Comment: The results show that as lag increases the subject's target recognition improves and the voice produces further changes on top of it." What "voice"? The auditory signal? Or: "This improvement exhibited a significant asymmetry across the audiovisual channel when compared after subtraction from the baseline condition." The improvement exhibited an asymmetry? What is subtracted from what? What is compared to what? The sentence is unintelligible.

Response: Thank you for your insightful opinion. We have revised and made the presentation more standardized as follows: "Thirty-five subjects were included in this study. The differences in visual and auditory recognition in an audiovisual task versus a single-channel task were compared using lag versus pitch as the independent variable. The results show that as lag increased, the subject's target recognition improved and the pitch produced further improvements. These improvements exhibited a significant asymmetry across the audiovisual channel. Our findings could contribute to better utilization of audiovisual integration resources to improve attentional transients and auditory recognition decline, which is useful in areas such as driving and education." (see pages 1, Abstract).

Introduction

  1. Comment: P2 L54ff: "However, in focusing on the auditory improvement of AB these studies failed to further explore the effects of auditory stimulus material representations on AB. However previous focuses on the sound improvement AB have overlooked the issue of sound characterisation." What is the contrast of the second "however"? Is this not repeating the same statement?

Response: Thank you for pointing out the error. We have removed the duplicated content. 

  1. Comment: In multiple places throughout the manuscript "tone" is used. What exactly is meant by "tone"? Is it not "pitch", which is also used at multiple places, but seemingly meaning something else than "tone"? For example, on P2 L58ff we read: "People perceive different tones differently. For example, in everyday life, people are more likely to be attracted to sharper sounds." What is a "sharp sound" in this context, and why are people "attracted" to it? This does not make much sense to me. And further we read: "In the classroom, the teacher may raise the pitch or make the voice lower when he/she is talking about the key points." A "lower" voice usually means a softer, quieter voice, but is this what the authors want to say here? They continue: "This shows that tone helps to improve visual perception." I do not see the connection here. Why does being "attracted" to a "sharper sound", or raising the pitch in the classroom support the conclusion that tone helps to "improve" visual perception? The authors go on: "Many studies have confirmed that looking at sound can improve vision." How could one possibly "look at sound"? And what type of "sound", is it music, or noise, or spoken words, and what is the connection to "tone", which is the actual topic of this paragraph?

Response: We thank the reviewer for the careful consideration. We have revised them as you suggested as follows:

Q1: "In multiple places throughout the manuscript "tone" is used. What exactly is meant by "tone"? Is it not "pitch", which is also used at multiple places, but seemingly meaning something else than "tone"? For example, on P2 L58ff we read: "People perceive different tones differently. For example, in everyday life, people are more likely to be attracted to sharper sounds." What is a "sharp sound" in this context, and why are people "attracted" to it? This does not make much sense to me."

Ans1: Sorry for the confusing writings. Here, tone refers to the quality or sound of a person's voice, while pitch refers to the highness or lowness of a sound. In this study, we would like to detect pitch actually. We have corrected them throughout the manuscript accordingly.

There are a number of language problems in the Introduction section. We have revised them with the help of professional language editors. It has been revised as follows"Pitch is an important element in the composition of sound. People perceive different pitches differently. For example, in everyday life, people are more likely to perceive higher pitched sounds; higher frequency sounds are perceived as brighter and sharper, and are therefore more likely to be noticed. In the classroom, teachers may raise the pitch to emphasize key knowledge in the textbook when they are talking about key content. These cases show that changes in pitch can help improve visual perception." (see pages 2, line 63).

Q2: "A "lower" voice usually means a softer, quieter voice, but is this what the authors want to say here?"

Ans: The use of "lower voice" is not relevant to the research in this paper and we have removed it.

Q3: "They continue: "This shows that tone helps to improve visual perception." I do not see the connection here."

Ans: Sorry for the confusing writings. We have revised it as follows:" higher frequency sounds are perceived as brighter and sharper, and are therefore more likely to be noticed."

Q4: "Why does being "attracted" to a "sharper sound", or raising the pitch in the classroom support the conclusion that tone helps to "improve" visual perception? "

Ans: "sharper sound" is mean "A higher pitched sound." . We have revised it into a more clear way as follows: " In the classroom, teachers may raise the pitch to emphasize key knowledge in the textbook when they are talking about key content [15]. These cases show that changes in pitch can help improve visual perception.".

Q5: "The authors go on: "Many studies have confirmed that looking at sound can improve vision." How could one possibly "look at sound"? And what type of "sound", is it music, or noise, or spoken words, and what is the connection to "tone", which is the actual topic of this paragraph? "

Ans: Sorry for the confusing writings. We have revised this paragraph into a more clear and precise way. It is as follows, "Sound can be used to improve vision. Pitch is an important element in the composition of sound." . There is a progressive relationship between sound and pitch.

  1. Comment: P2 L78ff: "Some studies have suggested that it is the uniqueness of the sound that improves AB, but when all stimuli are accompanied by sound AB does not differ from the accompanying sound at T2 only." AB is the phenomenon of attentional blink, which is in a totally different category than sound, so it is impossible that AB "differs" from the accompanying sound.

Response: Thank you for pointing out the mistake. We would like to explore the relationship between auditory and visual influences. Previous researches indicate that T2 is reinforced due to the accompaniment of sound only at T2, but there is no difference in the recognition of T2 when compared to a sound condition where all visual stimuli are accompanied by a sound [1, 2].

[1]. Yang, H., et al., Effect of Target Semantic Consistency in Different Sequence Positions and  Processing Modes on T2 Recognition: Integration and Suppression Based on  Cross-Modal Processing. Brain Sci, 2023. 13(2).

[2]. Aijun, W., et al., Modal-based attention modulates attentional blink. Attention, perception & psychophysics, 2022(2):

  1. Comment: P2 L91: "This leads to the phenomenon of asymmetry in audiovisual integration." What does "This" refer to?

 Response: We thank the reviewer for the careful consideration. We have amended the text to read. "Evaluations of audiovisual integration are usually based on conditions in which audiovisuals are presented simultaneously, as opposed to conditions in which only visuals or only audio stimuli are presented. When changes in visual recognition do not coincide with changes in auditory recognition, this produce audiovisual asymmetry. Several studies have shown that when visual and auditory stimuli are presented simultaneously and require discrimination between visual and auditory information, visual tasks are prioritized for debriefing and visual information tends to dominate the integration outcome."

  1. Comment: P2 L93ff: "In particular, the priority of visual information in audiovisual simultaneous judgement tasks [18]. The effect of visual quantitative information on audiovisual matching tasks [19]. Similarly affected by the time perception [11]." These are not full sentences and their meaning in the given context is unclear.6.P2 L93ff:"

Response: Thank you for pointing out the error. We have revised it as follow:"Several studies have shown that when visual and auditory stimuli are presented simultaneously and require discrimination between visual and auditory information, visual tasks are prioritized for debriefing and visual information tends to dominate the integration outcome [18]. A higher degree of reporting occurs when visual and auditory information is matched [19]. Similarly different time-invariant conditions also affect visual and auditory discrimination [11]. " (see pages 2-3, line 97).

  1. Comment: P2 L98: "perceptual perception" is a funny tautology that seems out of place here.

Response: Sorry for the mistake. We have deleted "perceptual".

  1. Comment: P3 L109f: "the improvement of sound for AB is due to the occupancy of auditory resources during recognition or recognition." It is certainly not the sound that is improved, but rather the attentional blink (AB) is improved, caused by the sound being delivered.

Response: Thank you for pointing out the error. We have revised it as follows: "AB is improved under audiovisual conditions which was caused by the sound being delivered " (see pages 3, line 118).

Methods:

  1. Comment: Where do the authors get the estimated effect size of F=0.3?

Response: Thank you for your question. With an estimated effect size f = 0.25, alpha = 0.05, and power = 0.9, G*Power suggested a sample size of 30 participants. We have revised the manuscript accordingly (see pages 3, line 126).

  1. Comment: It is reported that "Two subjects failed to discriminate between auditory stimuli and one subject was excluded for low accuracy. In other words, non-blinkers were excluded" Was the auditory accuracy determined beforehand or after examining the results? What exactly was the criterion of performing too poorly? And I do not see why people that perform poorly on discriminating auditory stimuli are "non-blinkers." Intuitively, a "non-blinker" would be someone that shows no attentional blink, which, however, seems unrelated to not being able to distinguish between stimuli. 

Response: Thank you for pointing out the error. We have revised the manuscript as you suggested as follows: "Two of the subjects were unable to accurately discriminate sound stimuli and were excluded, i.e., they provided too few correct auditory target reports. The other subject was excluded for providing too few correct visual target reports and was a non-blinker in visual target recognition." The criterion for too poor performance was that the subject was less than 30% correct, and the excluded subject was only 11% correct in T1. For T2 only 9% correct in lag3, and 8% correct in lag8, which is consistent with the definitional category of non-blinker when we consider that he is unable to accurately identify T1 and T2. The term "non-blinker" specifically refers to individuals who do not demonstrate the attentional blink effect, but it does not necessarily imply consistently high accuracy in processing T2. The reasons for variability in accuracy among non-blinkers are likely multifaceted and may be influenced by individual differences in attentional control, cognitive processing, and neural mechanisms.

  1. Comment: P3 L138: "collat" should mean "collected". P3 L144: "was used for T1" should be "were used for T1".

Response: Thank you for pointing out the error. We have corrected them as you suggested.

  1. Comment: P3 L147ff: What does "si" mean? E.g. "494hz, si in C major", or "988hz, si in C ascending". And it is impossible that a single pitch can be related to an entire chord (C major, and what is "C ascending", do they mean "C augmented"?). A chord consists of at least three differently pitched notes. If there are only two notes, we have an interval, and a single note is just a single note.

Response: Thank you for pointing out the error. The "si" here means "ti". There is a linguistic difference between Chinese and English. In Chinese, "ti" is pronounced "si" in Chinese. We have revised the manuscript as follows: "The auditory material consisted of pure pitches of ti tones in different keys, with an octave difference between the bass (494 hz, the fourth octave of B major), alto (988 hz, the fifth octave of B major), and treble (1976 hz, the sixth octave of B major)." (see pages 4, line 158).

  1. Comment: Experimental procedure: Why is there a presentation of "blank" in the visual stimuli, and a presentation of "no sound"?

Response: Thank you for your question. Referencing to Zhao, S., et al.'s study , the presence of visual interference stimuli from non-visual targets in the Only_A condition would also produce visual prioritization phenomenon and thus audio-visual integration would occur. Thus, "blank" was used to avoid the visual influence on hearing.

[1]. Zhao, S., et al., Attentional blink suppresses both stimulus‐driven and representation‐driven cross‐modal spread of attention. Psychophysiology, 2021.

[2]. Zhao, S., et al., The interplay between audiovisual temporal synchrony and semantic congruency in the cross-modal boost of the visual target discrimination during the attentional blink. Human brain mapping, 2022. 43(8): p. 2478-2494.

[3]. Zhao, S., et al., Updating the dual‐mechanism model for cross‐sensory attentional spreading: The influence of space‐based visual selective attention. Human Brain Mapping, 2021. 42(18): p. 6038-6052.

  1. Comment: Table 1: What is "Trail"? What are "Corresponding letters of the report"? What is meant by the statement: "It is possible to press the SPACE BAR, but correctness is not recorded"?  

Response: Thank you for your question. We have revised the diagram accordingly.  

"Trail" means "Visual and audio conditions". "Corresponding letters of the report"  means that "report the corresponding letter on the keyboard". "It is possible to press the SPACE BAR, but correctness is not recorded" means that in the Only_V vs. Only_A task, subjects could report another stimulus that was not present by pressing space, avoiding that keystroke habituation affected the perception of the stimulus.

  1. Comment: P4 L162ff: "[...] 23 visual stimuli were presented sequentially on the screen. Each stimulus was presented for 100 ms, with 50 ms of visual stimulus presentation followed by 50 ms of blank screen presentation." This does not make sense to me. There are 23 visual(!) stimuli that are followed by (another?) visual stimulus and a blank screen (which is, for that matter, also a visual stimulus)? In the next sentence, what can possibly be meant by "changes in visual stimuli were not obvious during visual perception"? And how can all these statements be reconciled with the statement that "When the presentation of the first stimulus ended, the next stimulus was presented immediately"? What happened to the blank screen in between? And how can this in turn be reconciled with "In this study the stimulus onset asynchrony (SOA) was only 100 ms"? Isn't it that the SOA is usually a variable time interval between the onset of subsequent stimuli? Here, it is fixed to 100 ms? And if so, how to understand the next sentence: "This is represented by the lag, which is the time interval of T2 relative to T1 when the SOA of T1 and T2 is 300 ms, or lag3"? And right after this sentence we learn that "Similarly, the other condition is lag8, which is the time interval of T2 relative to T1". But before it was lag3, not lag8? And after that we read the incomplete sentence: "21 number interference stimuli with 2 letter targets (T1, T2)." I am completely lost at this point.

Response: Thank you for your questions. We have revised the manuscript as follows: During an RSVP stream, 23 visual stimuli were presented sequentially on the screen. Each RSVP stream consisted of 21 number interference stimuli and 2 letter targets (T1, T2). Each stimulus consisted of a 50 ms presentation of the visual object (number or letter) with a 50 ms presentation of a blank screen. The blank screen was presented to avoid similarities when consecutively presented visual stimuli from influencing visual judgments. For example, 3 and 8 would produce similar percepts when presented consecutively [11]. T1 appeared randomly at positions 6 through 13 in the RSVP stream to prevent the subjects from producing rhythms. Lag denotes the lag position of T2 relative to T1, i.e., lag3 indicates that T2 appeared at the 3rd stimulus position after the presentation of T1, and lag8 indicates that T2 appeared at the 8th stimulus position after the presentation of T1. (see pages 4-5, line 179).

  1. Comment: When the authors speak of "reports" do they possibly mean "responses"?

Response: Thank you for pointing out the error. It means that "report the corresponding letter on the keyboard ". We have revised the manuscript accordingly.

  1. Comment: How can it be that "subjects were required to respond accurately during the response" (P4 L175)?

Response: Thank you for pointing out the error. We have revised it as follows: "subjects were asked to respond as accurately as possible during the task." (see pages 5, line 195).

  1. Comment:What is "T2 correctness with T1 correctness (T2T1), T3 correctness with T2T1 correctness (T3T2), ..."?

Response: Thank you for your question. T2T1 denotes the number of times Number of times subjects responded correctly to T2 when T1 responded correctly correctly. In an RSVP stream only if both T2 and T1 were reported correctly. T3T2 denotes the number of times T3 was correct in an RSVP stream only if both T3 and T2T1 were reported correctly. T3T1 denotes the number of times T3 correctness with T1 correctness when T2 did not present a letter (T3T1). Correctness when T2 did not present a letter (T3T1). (see pages 5, line 203).

  1. Comment: P5 L189: "Effect of audiovisual presentation on vision (AVPV) = T2T1 in audiovisual condition (VA_T2T1) - T2T1 in visual-only condition (only_V_T2T1). " What do these equations and abbreviations represent? These are all incomplete sentences with no apparent meaning!

Response: Thank you for your insightful opinion. We have revised it as follows: "The audiovisual asymmetry is reflected by the change of Effect of audiovisual presentation on vision (AVPV) and Effect of audiovisual presentation on audition (AVPA). AVPV is calculated by subtracting T2T1 in audiovisual condition (VA_T2T1) from T2T1 in visual-only condition (Only_V_T2T1) and AVPA is calculated by subtracting T3 in audiovisual condition (T3T2) from T3 in visual only condition (T3T1). This is calculated as follows:

AVPV = VA_T2T1 - Only_V_T2T1.

AVPA = T3T2 - T3T1."(see pages 5-6, line 210).

The content is referenced from the study of Zhao, S., et al [1,2]. By subtracting the method of comparing the responses, this helps to further verify the mechanism and mechanism of its generation through EEG experiments.

[1]. Zhao, S., et al., Attentional blink suppresses both stimulus‐driven and representation‐driven cross‐modal spread of attention. Psychophysiology, 2021.

[2]. Zhao, S., et al., Updating the dual‐mechanism model for cross‐sensory attentional spreading: The influence of space‐based visual selective attention. Human Brain Mapping, 2021. 42(18): p. 6038-6052.

Results:

  1. Comment: P5 L195f: "A two-way repeated measures ANOVA was conducted with T2T1 in the audiovisual condition as the dependent variable." This does not belong in the Results section but definitely in the Methiods section. And it has to be a bit more elaborate there. What are the two dependent variables (factors) of the two-way ANOVA? Have the requirements for an ANOVA been checked? Why is it a repeated-measures ANOVA and what is the temporal factor in this context?

Response: Thank you for your insightful opinion. 

Q1: "A two-way repeated measures ANOVA was conducted with T2T1 in the audiovisual condition as the dependent variable." This does not belong in the Results section but definitely in the Methiods section. And it has to be a bit more elaborate there.

Ans: We have revised it as you suggested in Methods section (2.4. Data Analysis).

Q2: "What are the two dependent variables (factors) of the two-way ANOVA?"

Ans: We conducted a two-way repeated measures analysis of variance with lag and pitch as independent variables and T2T1, T3T1, AVPV, and AVPA as dependent variables.

Q3: "has the requirements for an ANOVA been checked? "

Ans: We have checked the requirements for an ANOVA, including Independent and identically distributed (IID) samples,Normally distributed data,Homogeneity of variances,Interval or ratio scale data and Random sampling. The data meets the assumptions for ANOVA.

Q4: "Why is it a repeated-measures ANOVA and what is the temporal factor in this context?"

Ans: " the temporal factor" means "lag". There are several reasons that repeated measurements were used in this study. Firstly, a repeated-measures ANOVA are used when the same subjects are measured at multiple time points or under multiple conditions. This type of ANOVA is also known as within-subjects ANOVA. Secondly, the repeated-measures ANOVA are appropriate in situations where the data involve correlated measurements, such as when the same individuals are tested under different conditions, or when measurements are taken at multiple time points from the same individuals. This approach allows for the examination of within-subject effects while controlling for individual differences, making it a powerful tool for analyzing data with a repeated-measures design. The use of repeated-measures ANOVA can also  increased statistical power, as it can account for individual differences and reduce variability in the data.

We have revised the manuscript accordingly as follows. "All data were subjected to repeated measures ANOVA after satisfying the tests for sample independence, normality, and chi-square test. We conducted a two-way repeated measures analysis of variance with lag and pitch as independent variables and T2T1, T3T1, AVPV, and AVPA as dependent variables. In the presence of interaction effects, a post hoc analysis using the Least Significant Difference (LSD) method was performed for multiple comparisons." (see pages 6, line 218).

  1. Comment:If and when there is a significant interaction effect between factors, the corresponding main effects must be taken with caution, and may be even meaningless. There is no discussion of this issue.

Response: Thank you for pointing out error. Previously we would like to emphasize that lag3 is less than lag8 in terms of correctness of subjects' reports, and that there is still AB. However, it may be inappropriate as you suggested. We have revised the manuscript accordingly. (see pages 6, line 227).

  1. Comment:"lag3 was significantly lower than lag8" Not the conditions themselves were lower but the subject's performance under these conditions.

Response: Thank you for pointing out error. We have revised the manuscript as follows. "the subjects in the lag3 condition gave significantly fewer correct answers than in the lag8 condition." (see pages 6, line 231).

  1. Comment: Why is there (apparently) no post-hoc comparison after the ANOVA of the T2T1 lag?

Response: Thank you for pointing out error. We have made the post hoc comparison and added the results into the manuscript. "Post hoc comparisons showed that in the lag8 condition, the subjects’ recognition of T2 with the treble accompaniment was lower than with the bass accompaniment (p = 0.031, Cohen’s d = -0.078, 95% CI [-0.045, -0.002]) and no sound accompaniment (p = 0.017, Cohen’s d = -0.099, 95% CI [-0.057, -0.006]) " (see pages 6, line 238).

Figure 2:

  1. Comment: What do the connection lines between the values represent? How come they are even curved in a rather weird way? They are indicated in the legend as "Tend lines". Do the authors mean "trend lines"? But even so, they are drawn between distinct categories, so there actually is no meaning to values in between these categories.

Response: Thank you for pointing out error. We have added trend lines to reflect that AB benefits decrease as the pitch rises. We have revised Figure 2 accordingly (see pages 6, Figure 2).

  1. Comment: What do the error bars represent?

Response: The error bars represent the standard errors. We have revised Figure 2 accordingly (see pages 6, Figure 2).

  1. Comment: I assume the asterisks between two values with error bars on top of each other indicate a significant difference. Why is there no significant difference between the last two values, even though the errors bars do not overlap?

Response: Sorry for the mistake. We have revised Figure 2 accordingly (see pages 6, Figure 2).

  1. Comment: Why did the authors choose such a rather unorthodox graphic representation and not a representation as in Figure 3? The data have a similar structure.

Response: Thank you for your insightful opinion. At that place we would like to show the gradual decrease in AB benefit as the pitch rises. We have added another figure to express it more clearly. (see pages 6, Figure 2b).

  1. Comment: P6 L220f: "Tonal stimuli, on the other hand, showed relatively centred sounds that was approximately difficult to discriminate." What does this sentence mean?

Response: Sorry, this was a linguistic error. We would like to express that the alto condition is less correct. We have revised the paragraph accordingly as follows. "Auditory discrimination is affected by lag and pitch. Specifically, the larger the lag the better its auditory discrimination. For pitch, the alto condition showed a lower level of recognition, while the treble vs. bass condition showed a higher level of recognition." (see pages 7, line 255).

  1. Comment: Title of subsection 3.3: "Comparison of the amount of change in T2 and T3." What change is meant here?

Response: Thank you for your question. We would like to compare the difference between Effect of audiovisual presentation on vision (AVPV) and Effect of audiovisual presentation on audition (AVPA).  (see pages 7, line 260).

  1. Comment: Also the description of the two-way repeated measures ANOVA belongs in the Methods section, with an appropriate and more elaborate explanation.

Response: Thank you for your insightful opinion.. We have revised it as the Methiods section. And it has to be a bit more elaborate there.(see pages 6, line 218).

General criticism:

  1. Comment: The fact that the auditory stimulus is presented at the same time as the second visual stimulus makes the discrimination tasks with respect to these two stimuli of different modality a dual-task scenario. Thus, there is a shared attention between the two discrimination tasks, which hinders a clear interpretation with respect to the actual aim of the study: to find out to what extent sound improves visual discrimination. In the experimental setting designed by the authors, the auditory stimulus draws attention away from the visual discrimination task, because the subject has to discriminate not only the visual stimuli but also the auditory stimuli. While this is interested in itself, the discussion of this crucial point is very sparse and confusing. For example, on P8 L300, the authors write: "The stimulus representations of sound did not produce an interaction with lag. This suggests that their effects on attention are independent of each other." On the one hand, this is a very strong conclusion that is not in line with established views. Secondly, the authors found that "The sound and lag interaction effect was significant" (P5 L198f), which seems to directly contradict their statement that the stimulus representations of sound did not produce an interaction with the lag between the visual stimuli.

Response: Thank you for your insightful opinion.

Q1: "Thus, there is a shared attention between the two discrimination tasks, which hinders a clear interpretation with respect to the actual aim of the study: to find out to what extent sound improves visual discrimination. "

Ans: In the position of T2, simultaneous perception of vision and hearing in the T2 position is a dual task. It is not possible to conclude whether shared attention exists across audiovisual channels. This is because it is not part of the alerting and orienting part of attention (Zhao, S., et al., 2021; Yang, H., et al., 2023), but it may be present in the higher processing stages of attention. (see pages 10, line 361)

Q2: "In the experimental setting designed by the authors, the auditory stimulus draws attention away from the visual discrimination task, because the subject has to discriminate not only the visual stimuli but also the auditory stimuli. "

Ans: In the results of AVPV it can be seen that the recognition of vision is instead improved in the dual task. This has been interpreted in some studies as a result of audiovisual integration (Aijun, W., et al., 2022). There is some debate about this point of inquiry, starting with the fact that some studies consider the process of audiovisual integration to be automated (Jinglong, et al., 2016). However, the underlying premise of automation is that it does not require advanced processing (Jinglong, et al., 2016). Prior studies in experiments designed for stimulus conditions that are typically audiovisually synchronized do not need to report auditory only visual. Thus perceiving auditory stimuli may exist only at the primary processing stage. Subsequent studies have found that processing modality similarly affects audiovisual integration (Jinglong, et al., 2016; Yang, H., et al., 2023), but whether it enhances or inhibits it is not currently standardized. In the case of positive AVPV results, it was argued that the "processing mode" or "the auditory stimulus draws attention away from the visual discrimination task" design also has an impact on audiovisual integration (Jinglong, et al., 2016; Yang, H., et al., 2023). The "processing mode" or "the auditory stimulus draws attention away from the visual discrimination task" design also has audiovisual integration, i.e., the dual task promotes visual discrimination equally. (see pages 10, line 365)

Q3: "which hinders a clear interpretation with respect to the actual aim of the study: to find out to what extent sound improves visual discrimination. "

Ans: The above strong conclusions help us to continue to explore "to what extent sound improves visual discrimination", which we found by controlling for pitch, and which seems to be related to the level of pitch (Figure 2a). Comparison of the difference between lag3 and lag8 shows that AB's seems to disappear as the pitch changes. However, in the actual statistics, there is a decreasing trend at lag8 and an increasing trend at lag3 as the pitch increases. This phenomenon we believe may be due to the fact that the total amount of cognitive resources is fixed. If this view is valid, it can be inferred that there is a possibility of mutual transformation between audiovisual resources, and that this transformation process is most likely influenced by sound factors. (see pages 10, line 376)

Q4: "The stimulus representations of sound did not produce an interaction with lag. This suggests that their effects on attention are independent of each other." and "The sound and lag interaction effect was significant."

Ans: These two points were misrepresented by us. We have refined it in our discussions. We think that comparing T2T1 and T3T2 in different lag and pitch conditions, it can be further inferred that the effect of hearing on vision is unidirectional, but vision and hearing is perceives in parallel. The stimulus representation of sound did not interact with lag, but the phenomenon present in visual recognition validates the prioritization of visual processing. (see pages 9, line 348)

    In summary, our study concluded that the phenomenon of audiovisual integration occurs whether or not it is a dual task. However, in the present study, an asymmetry emerged in the audiovisual integration phenomenon we explored. This asymmetry was manifested in two ways; on the one hand, there was an asymmetry in the audiovisual improvement at lag3. That is, the inconsistency between the auditory improvement and visual improvement. On the other hand, there is a negative auditory change at lag8. That is, the prioritization of vision is more pronounced at non-bottleneck locations.

[1] Zhao, S., et al., Updating the dual‐mechanism model for cross‐sensory attentional spreading: The influence of space‐based visual selective attention. Human Brain Mapping, 2021. 42(18): p. 6038-6052.

[2] Yang, H., et al., Effect of Target Semantic Consistency in Different Sequence Positions and  Processing Modes on T2 Recognition: Integration and Suppression Based on  Cross-Modal Processing. Brain Sci, 2023. 13(2).

[3] Aijun, W., et al., Modal-based attention modulates attentional blink. Attention, perception & psychophysics, 2022(2): p. 84.

[4] Jinglong, et al., The interactions of multisensory integration with endogenous and exogenous attention. Neuroscience and Biobehavioral Reviews, 2016.

Round 2

Reviewer 2 Report

Comments and Suggestions for Authors

The authors have addressed the issues raised in my previous report and have significantly improved the manuscript.  However, there are still some issues that I would like to be resolved:

P2 L55: "Sound can be used to improve vision" and P2 L67: "These cases show that changes in pitch can help improve visual perception."  These modified sentences and the underlying notion still seem wrong to me.  Raising the pitch does not actually improve visual perception.  Rather, it increases attention and therefore helps students to better receive and digest the information presented to them, be it in visual or auditory form.  After all, it is generally not impaired visual perception that prevents students from following a course, but rather lack of attention.  This is not the same thing.  The students do not actually see better, when the teacher raises their pitch.  They listen up and improve their processing of information.

P4 L153: "The auditory material consisted of pure pitches of ti tones in different keys [... ]" The authors have corrected their reference to "si tones" to "ti tones", but this does not make it any better.  What are ti tones?  I googled it but found no conclusive answer.

In the same place, why is 494 Hz (the unit of frequency, by the way, is "Hz" with a capital "H", not "hz")  "the fourth octave of B major" and "988 hz, the 154th octave of B major"?  I am a musician myself and have some knowledge of (Western) music theory, but this makes no sense to me.  As I said in my previous report, a chord consists of at least three notes.  Two notes form an interval, not a chord.  B major would consist of the notes B, D# and F#.  Chords do not depend on absolute pitch, so there is no such thing as a "fourth octave of B major", at least not in Western music theory.  If this makes sense with regards to Chinese music theory, it would be helpful to briefly describe it to make this part understandable to a wider audience.

In Figure 2a it is difficult to see the error bars because the lines overlap.  Would it be possible to move each pair of values, which are now shown exactly on top of each other, slightly away from each other (along the x-axis) so that they do not overlap?

Inspired by my criticism, Figure 2 now shows the same data in two different ways in panels a and b. Although this is mentioned in the main text, it is not clear from the caption and should be made explicit there.

In Figures 2-4, the meaning of the error bars should be indicated in the caption, e.g.  "Error bars represent the standard error."

Author Response

Reviewer 2

Thank you for the valuable comments. We have revised the manuscript carefully as you suggested.

1. Comment: P2 L55: "Sound can be used to improve vision" and P2 L67: "These cases show that changes in pitch can help improve visual perception."  These modified sentences and the underlying notion still seem wrong to me.  Raising the pitch does not actually improve visual perception.  Rather, it increases attention and therefore helps students to better receive and digest the information presented to them, be it in visual or auditory form.  After all, it is generally not impaired visual perception that prevents students from following a course, but rather lack of attention.  This is not the same thing.  The students do not actually see better, when the teacher raises their pitch.  They listen up and improve their processing of information.

Response: Thank you for pointing out the error. We have revised it as follow:sound can be used to improve attention and These cases show that changes in pitch can help improve attention. By improving attention it helps people to better receive and digest information presented to them in visual or auditory form.  (see pages 2, line 55 & 67).

2. Comment: P4 L153: "The auditory material consisted of pure pitches of ti tones in different keys [... ]" The authors have corrected their reference to "si tones" to "ti tones", but this does not make it any better.  What are ti tones?  I googled it but found no conclusive answer.In the same place, why is 494 Hz (the unit of frequency, by the way, is "Hz" with a capital "H", not "hz")  "the fourth octave of B major" and "988 hz, the 154th octave of B major"?  I am a musician myself and have some knowledge of (Western) music theory, but this makes no sense to me.  As I said in my previous report, a chord consists of at least three notes.  Two notes form an interval, not a chord.  B major would consist of the notes B, D# and F#.  Chords do not depend on absolute pitch, so there is no such thing as a "fourth octave of B major", at least not in Western music theory.  If this makes sense with regards to Chinese music theory, it would be helpful to briefly describe it to make this part understandable to a wider audience.

Response: Thank you for pointing out the error. We have revised it as follow:The auditory material consisted of pure pitches of B in different keys, with an octave difference between the bass B4 (494 Hz), alto B5 (988 Hz), and treble B6 (1976 Hz).  (see pages 4, line 158).

3. Comment:In Figure 2a it is difficult to see the error bars because the lines overlap.  Would it be possible to move each pair of values, which are now shown exactly on top of each other, slightly away from each other (along the x-axis) so that they do not overlap?

Response: Thank you for your insightful opinion. It has been redrawn (see pages 6, Figure 2). 

4. Comment:Inspired by my criticism, Figure 2 now shows the same data in two different ways in panels a and b. Although this is mentioned in the main text, it is not clear from the caption and should be made explicit there.

Response: Thank you for your insightful opinion. It has been indicated (see pages 6, Figure 2). 

5. Comment: In Figures 2-4, the meaning of the error bars should be indicated in the caption, e.g.  "Error bars represent the standard error."

Response: Thank you for your insightful opinion. It has been indicated (see pages 6-8, Figure 2-4).